# Molecular Epidemiology of Multidrug-Resistant Pneumococci among Ghanaian Children under Five Years Post PCV13 Using MLST

**DOI:** 10.3390/microorganisms10020469

**Published:** 2022-02-19

**Authors:** Richael O. Mills, Mohammed R. Abdullah, Samuel A. Akwetey, Dorcas C. Sappor, Gustavo Gámez, Sven Hammerschmidt

**Affiliations:** 1Department of Molecular Genetics and Infection Biology, Interfaculty Institute for Genetics and Functional Genomics, Center for Functional Genomics of Microbes, University of Greifswald, 17487 Greifswald, Germany; richael.mills@ucc.edu.gh (R.O.M.); mohammed.abdullah@med.uni-greifswald.de (M.R.A.); 2Department of Biomedical Sciences, School of Allied Health Sciences, University of Cape Coast, Cape Coast PMB TF0494, Ghana; sakwetey@uds.edu.gh; 3Department of Clinical Microbiology, School of Medicine, University of Development Studies, Tamale PMB TF0494, Ghana; 4Department of Medical Laboratory Sciences, School of Allied Health Sciences, University of Cape Coast, Cape Coast PMB TF0494, Ghana; dorcaschristiana20@gmail.com; 5Basic and Applied Microbiology (MICROBA) School of Microbiology, University of Antioquia Medellin, Medellin 050010, Colombia; gustavo.gamez@udea.edu.co

**Keywords:** multidrug-resistance, pneumococci, PCV13, resistance genes, children, Ghana

## Abstract

Antibiotic resistance in pneumococci contributes to the high pneumococcal deaths in children. We assessed the molecular characteristics of multidrug-resistant (MDR) pneumococci isolated from healthy vaccinated children under five years of age in Cape Coast, Ghana. A total of 43 MDR isolates were selected from 151 pneumococcal strains obtained from nasopharyngeal carriage. All isolates were previously serotyped by multiplex PCR and Quellung reaction. Susceptibility testing was performed using either the E-test or disk diffusion method. Virulence and antibiotic resistance genes were identified by PCR. Molecular epidemiology was analyzed using multilocus sequence typing (MLST). Vaccine-serotypes 23F and 19F were predominant. The *lytA* and *pavB* virulence genes were present in all isolates, whiles 14–86% of the isolates carried pilus-islets 1 and 2, *pcpA*, and *psrP* genes. Penicillin, tetracycline, and cotrimoxazole resistance were evident in >90% of the isolates. The *ermB*, *mefA*, and *tetM* genes were detected in (n = 7, 16.3%), (n = 4, 9.3%) and (n = 43, 100%) of the isolates, respectively. However, >60% showed alteration in the *pbp2b* gene. MLST revealed five novel and six known sequence types (STs). ST156 (Spain^9V^-3) and ST802 were identified as international antibiotic-resistant clones. The emergence of international-MDR clones in Ghana requires continuous monitoring of the pneumococcus through a robust surveillance system.

## 1. Introduction

*Streptococcus pneumoniae* (pneumococcus) belongs to the normal flora in the nasopharynx and has been implicated in invasive diseases such as pneumonia, sepsis, and meningitis. In other situations, pneumococci can cause mild infections such as sinusitis and otitis media. The global burden of pneumococcal disease is enormous and mostly borne by children under five years, the elderly, and persons with a compromised immune system. As such, the pneumococcus is included in the nine bacterial pathogens of international concern mainly due to the detection of antibiotic resistance, particularly penicillin-non-susceptibility [1].

The highest burden of invasive pneumococcal disease (IPD) and deaths is found in Africa and Asia [2]. To reduce the global burden of pneumococcal morbidity and mortality, pneumococcal vaccines recommended by the World Health Organization (WHO) have been introduced in several countries including Ghana [2]. Furthermore, results of a global modeling analysis on the cost-effectiveness of introducing pneumococcal vaccines in 180 countries showed that 34% of global pneumococcal deaths (0.399 million (0.208 million to 0.711 million)) and 12% of pneumococcal disease episodes (54.6 million (51.8 million to 58.6 million)) could be averted [3]. Hence, in 2012 Ghana introduced the 13-valent pneumococcal conjugate vaccine (PCV13) into the childhood immunization program. Although PCVs have reduced IPD, they have also led to the emergence of non-vaccine serotypes [4,5]. Therefore, protein-based vaccines have been proposed as potential-alternative to polysaccharide-based vaccines [6]. Some highly conserved proteins in pneumococci including lytic amidase A (LytA), pneumolysin (Ply), and pneumococcal choline-binding protein A (PcpA) have been identified to be immunogenic. Other proteins located in pneumococcal pathogenicity islands such as pneumococcal serine-rich repeat protein (PsrP) and the pilus islets have also been studied. These proteins are currently at different stages of clinical trials [6].

Antibiotics have been used in the treatment and management of pneumococcal diseases for a very long time. However, the increase in global consumption of antibiotics and the highly competent nature of *S. pneumoniae* fostering uptake of genes from its environment has led to the observed high non-susceptibility to various antibiotics in some pneumococcal populations [4]. Furthermore, pneumococcal resistance patterns vary among different geographical populations and the global emergence of multidrug resistance (MDR) in non-vaccine serotypes (NVTs) has been reported [4,5,7]. Comparing pre and post PCV13 pneumococcal data from Ghana showed a decline in MDR ranging from 39.7–87% to 20.5–28.4% [5,8,9,10]. Moreover, previous data from Ghana shows a high resistance rate to commonly used antibiotics such as penicillin, tetracycline, and cotrimoxazole with a gradual increase in erythromycin resistance [11].

Structural alterations in the transpeptidase domain of three (*pbp 1a*, *2x*, *and 2b*) out of the six penicillin-binding proteins (pbp) have formed the genetic basis for penicillin resistance in pneumococci, while target modification and active efflux pump mechanisms facilitate macrolide resistance in *S. pneumoniae.* Target modification is mediated by the acquisition of the *ermB* (erythromycin-resistance methylase) gene. A methylase encoded by the *ermB* gene methylates 23S rRNA to induce a conformational change in the ribosomal target site, thereby impairing antibiotic binding. The *ermB* gene also mediates resistance in lincosamides (e.g., clindamycin and streptogramin B), resulting in a phenotype known as MLSB (macrolide, lincosamide, streptogramin B type). This phenotype is responsible for high-level resistance in pneumococci [4]. Furthermore, transposons belonging to the *Tn916* family can carry the *ermB* gene in their genetic composition and insert them into pneumococcal genomes [12]. Unlike *ermB*-mediated resistance, which is common in Asia, Africa, and Europe, efflux mediated resistance mostly involving the *mefA/E* genes have been associated with UK, North, and Latin America [13]. Meanwhile, the acquisition of the popular *tetM* and the rarely occurring *tetO* genes mediate tetracycline resistance through transposon activity and protection of the 30s ribosome [13]. The *Tn916-Tn1545* transposon family can carry both *tetM* and *ermB* genes and hence, recent studies have cited the possible occurrence of erythromycin and tetracycline co-resistance [14].

With the increasing trend in MDR among pneumococcal strains globally, the pneumococcal molecular epidemiology network (PMEN) and the PubMLST databases provide a platform for comparison of multilocus sequence type (MLST) strains with other globally disseminated MDR strains. Previous studies in Ghana have identified the proliferation and dissemination of lineages of ST802, ST9090, ST4194, and the hypervirulent ST217 among both carriage and invasive pneumococcal isolates [9,15,16,17].

One of the expected benefits of PCV introduction is to minimize the severity of invasive pneumococcal disease and mortality. PCVs, as with other vaccines, confer immunity before an infection could develop after exposure to a pathogen. Hence, PCV vaccination can reduce the carriage of antibiotic-resistant serotypes thereby limiting spread of the infection among the population. When this happens, antibiotic use for the treatment of pneumococcal infections will be limited leading to a general decline in antibiotic resistance. However, previous studies have shown that the increasing antibiotic resistance in emerging NVTs could defeat this expectation. Recent carriage studies from Ghana have shown a decline in penicillin resistance among PCV13 serotypes, but with a corresponding increasing resistance in non-PCV13 serotypes [5,8]. These studies from Ghana supported by recent global antibiotic resistance meta-data analysis showed no significant impact of PCV on tetracycline, erythromycin, and cotrimoxazole resistance [7]. However, there are limited data on the determinants of the genetic basis of pneumococcal antibiotic-resistant isolates in Ghana [8,18]. Again, the extent to which these MDR strains relate with internationally disseminated resistant clones is poorly understood. Therefore, this study aimed to investigate the phenotypes, genotypes, and genetic relatedness of multidrug-resistant pneumococcal carriage isolates. Furthermore, the prevalence of virulence factors was also determined.

## 2. Materials and Methods

### 2.1. Pneumococcal Isolates and Serotyping

All of the *S. pneumoniae* isolates were obtained from the nasopharynx of healthy vaccinated children recruited from kindergartens and immunization centers in Cape Coast, Ghana in 2018. The microbiological isolation, identification, and epidemiological information of the pneumococcal strains have been studied previously [5]. In brief, presumptive pneumococcal isolates were identified using the optochin and bile solubility tests. The pneumococcal isolates were serotyped by both multiplex PCR and the Quellung reaction.

### 2.2. Antibiotic Susceptibility Testing

The Kirby–Bauer disk diffusion method was used to assess susceptibility patterns of the pneumococcal strains to selected antibiotics. The antibiotics (Thermo Fisher, Dreieich, Germany) included levofloxacin (5 μg), vancomycin (30 μg), linezolid (30 μg), clindamycin (2 μg), erythromycin (15 μg), tetracycline (30 μg), chloramphenicol (30 μg), cotrimoxazole (25 μg), and ceftriaxone (30 μg). Oxacillin (1 μg) disks were used to screen for penicillin non-susceptibility. In brief, pneumococcal isolates were inoculated onto Mueller-Hinton agar supplemented with 5% sheep blood (Oxoid, Wesel, Germany) after which the antibiotic disks were applied. The agar plates were incubated overnight for 18–24 h in 5% CO_2_. Each test batch was incubated with *S. pneumoniae* ATCC 49619, which was used as a control strain.

Minimum inhibitory concentration (MIC) for penicillin was determined using E-test strips (Liofilchem, Roseto degli Abruzzi, Italy) for all oxacillin-resistant pneumococcal isolates. Isolates with MIC ≤ 0.06 μg/mL, 0.12–1.0 μg/mL, and ≥2.0 μg/mL were defined as penicillin-susceptible, penicillin-non-susceptible, and penicillin-resistant pneumococci, respectively.

All erythromycin-resistant isolates were further tested for inducible clindamycin resistance (D-zone test). Briefly, a 15 μg erythromycin disk was placed 12 mm away from a 2 μg clindamycin disk on Mueller-Hinton agar supplemented with 5% sheep blood and incubated overnight for 20–24 h in 5% CO_2_. A positive result of inducible clindamycin resistance was reported when a flattened zone of inhibition adjacent to the erythromycin disk was observed. Pneumococcal isolates that were resistant to three or more different classes of antibiotics were classified as multidrug-resistant (MDR). All susceptibility tests and result interpretations were performed according to the guidelines and criteria established by the Clinical and Laboratory Standard Institute (CLSI) [19].

### 2.3. DNA Extraction of MDR Isolates

Genomic DNA was extracted from pneumococcal cells using QIAamp DNA Mini Kit (Qiagen, Hilden, Germany). The extracted DNA for each isolate was used as template for all molecular tests as described [5].

### 2.4. PCR Amplification of Resistance Genes

Penicillin, erythromycin, and tetracycline resistance genes were amplified by PCR. Previously described primers [20] targeting the altered gene region of the *pbp2b* gene were used to detect the genetic basis for penicillin resistance among our MDR strains. Macrolide resistance genes including *ermB*, *mefA*, and *mefE* genes were amplified using published primers [21,22]. The *tetM* gene primers described by Malhotra-Kumar and colleagues were used in a conventional PCR to determine tetracycline resistance among the isolates [23]. Amplified DNA was visualized on 0.8% agarose gel and images were taken using the Intas^®^ gel documentation system.

### 2.5. Virulence Genes Determination

The detection of pneumococcal virulence genes including *cpsA*, *lytA*, *pavB*, *pcpA*, *psrP*, the pilus islets (PI) *PI-1*, and *PI-2* by PCR have been described earlier [5].

### 2.6. Multilocus Sequence Typing (MLST)

The MLST protocol for *S. pneumoniae* prescribed in the PubMLST [24] database was used to determine the sequence types (STs) among 20 isolates resistant to >4 different classes of antibiotics. We used a combination of alternative primers published by the CDC [25] and primers described in the PubMLST database to amplify the internal fragments of seven housekeeping genes namely *aroE*, *gdh*, *gki*, *recP*, *spi*, *xpt* and *ddl*. A 50 μL reaction mixture was prepared with 1 μL of 100 ng of DNA template, 1 μL of 25 mM MgCl_2_ (Roth), 1 μL of 5 mM dNTPS (Thermofisher), 5 μL of 10× Dream buffer (Thermofisher), 1 μL of the respective primers, and 0.5 μL of DreamTaq DNA polymerase (Thermofisher), and nuclease-free water. The following thermocycling conditions were used: 4 min hold at 94 °C, followed by 30 cycles of 95 °C for 30 s, 55 °C for 30 s, and 72 °C for 60 s, and a final extension at 72 °C for 5 min.

The amplified internal fragments were sequenced on both strands using the primers that were previously used for the amplification. Sequencing was performed at Macrogen, Inc. (Amsterdam, The Netherlands), while sequencing analysis was carried out with DNADynamo (Blue Tractor Software Ltd., Llanfairfechan, Wales). The consensus sequences were queried in the PubMLST database. Alleles that match an existing allele type number were assigned to those existing numbers. Sequences that did not match any existing allele type number were assigned new allele type and ST numbers. The genetic relatedness of the STs was analyzed using the goeBURST program [26]. Cluster analysis was performed using the PHYLOViZ program to generate minimum spanning and neighbour joining trees. Furthermore, STs were compared with data in the Pneumococcal Molecular Epidemiology Network (PMEN) database [27].

### 2.7. Statistical Analysis

Statistical analysis was performed using Statistical Package for Social Sciences (SPSS) software^®^ (version 21) and GraphPad Prism (GraphPad software^®^) version 5. Significance comparison for categorical data was expressed as proportions and tested with Chi-square test or Fisher’s exact test (two-tailed). *p* < 0.05 was considered to be statistically significant.

### 2.8. Ethical Approval

The institutional review board of the University of Cape Coast granted ethical approval for this study (UCCIRB/EXT/2017/21). In addition to parental consent, all the children voluntarily participated in this study.

## 3. Results

### 3.1. Brief Description of Isolates

One hundred and fifty-one pneumococcal isolates were originally obtained from 513 children ≤5 years of age, of which 43 were MDR. These 43 MDR isolates were obtained from 17 (39.5%) and 26 (60.5%) males and females, respectively. The highest proportion of MDR carriage was seen in children aged 2 and 3 years with 12 (27.9%) isolates each. Eleven different serotypes were observed with serotypes 23B, 23F, and 19F being the top three (Table 1). PCV13 covered 63.6% of the identified serotypes.

### 3.2. Antibiotic Resistance Pattern

All of the 43 isolates were fully susceptible to vancomycin, levofloxacin, and linezolid. However, 2 (4.7%), 11 (25.6%), 8 (18.6%), and 11(25.6%) of the isolates were non-susceptible to ceftriaxone, erythromycin, clindamycin, and chloramphenicol, respectively. Inducible clindamycin resistance was not observed among the study isolates. Marked resistance to penicillin 40 (93%), of which 39 and 1 isolates showed intermediate resistance and full resistance, respectively, to cotrimoxazole 42 (97.7%), and tetracycline 43 (100%) were observed in this study.

### 3.3. Prevalence of Resistance Genes in MDR Isolates

PCR analysis showed a relationship between the phenotypic and genotypic expression of antibiotic resistance. Of the 11 erythromycin-resistant isolates, 7 (63.6%) and 4 (36.4%) harbored the *ermB* and *mefA* genes, respectively (Table 1). However, the *mefE* gene was not detected in this study. Among the 40 penicillin-resistant isolates, 26 (65%) showed alterations in the *pbp2b* gene. All of the 43 tetracycline-resistant isolates tested positive for the *tetM* gene. Whereas the *tetM* and *pbp2b* genes were fairly distributed among the isolates, the *mefA* and *ermB* genes were limited to serotypes 9V, 19F, 38, and 6B, 19F, 23B, and 35B, respectively (Table 1).

### 3.4. Prevalence of Virulence Genes

The *cpsA*, *lytA*, *and pavB* virulence genes were highly conserved among the isolates and were therefore detected in all of the 43 tested MDR isolates. However, the *pcpA* and *psrP* genes were detected in 37 (86%) and 17 (39.5%) of the isolates, respectively (Figure 1). The pilus islets, *PI-1*, was limited to serotypes 19F, 6B, and 9V accounting for eight (18.6%), while *PI-1* was only detected in four serotype 19F isolates (14%). The possession of both *PI-1* and *PI-2* was observed in four out of six serotype 19F isolates.

### 3.5. Genetic Relatedness of MDR Isolates

A total of 20 pneumococcal isolates resistant to ≥4 different classes of antibiotics were analyzed using MLST. The analysis revealed a total of 11 STs, of which 5 were novel and 6 already known. Fourteen isolates (serotypes 9V, 19F, 23F, 35B, and 14) belonged to known STs, while six isolates (serotype 19F, 6B, and 23B) were clustered within three novel STs (Table 2). The most common ST was ST802 with seven isolates, of which six were serotype 23F and one serotype 19F. Figure 2 shows the relatedness of the different serotypes with the identified STs.

The *pcpA* gene was present in all STs except for ST983. However, the isolate belonging to ST983 was positive for *psrP*, *PI-1*, and *PI-2* (Figure 3). The *PI-1* and *PI-2* seen in ST802, ST15458, and ST15459 were contributed to by serotype19F strains.

STs 983 and 15448 were triple locus variants (TLVs) of ST236 (Taiwan^19F^-14). STs 15111 and 15461 were double locus variants of ST338 (Columbia^23F^-26) and ST273 (Greece^6B^-22), respectively.

The genetic relatedness between our strain collection and that of isolates compiled in the Pneumococcal Molecular Epidemiology Network (PMEN) is shown in Figure 4. Two of our serotype 9V isolates were associated with PMEN clone ST156 (Spain^9V^-3).

## 4. Discussion

The present study is focusing on the molecular characteristics of multidrug-resistant pneumococcal isolates. Susceptibility to ceftriaxone, vancomycin, levofloxacin, and linezolid was very high, which is in accordance to observations in previous studies [10,28,29,30]. The high susceptibility rate to ceftriaxone a drug of choice for treating invasive pneumococcal disease in Ghana is consistent with both carriage and outbreak data [9,16,30]. This gives an indication that ceftriaxone can still be used for the effective treatment and management of pneumococcal infections in Ghana.

The use of cotrimoxazole as a prophylactic in HIV/AIDs infection, in the treatment of respiratory infections in resource-poor countries, coupled with its ease of accessibility as an over-the-counter-drug, has been shown to contribute to the high resistance to cotrimoxazole in many African countries [31]. The trend of high resistance to cotrimoxazole across the different serotypes observed in this study is similar to findings from other African countries [30,32]. Among the different antibiotics tested, resistance to tetracycline was persistently high (100%), which is consistent with earlier reports [8,12,28]. In Ghana, persistently high resistance to tetracycline has been seen in pneumococci and other bacteria [11]. Factors fueling the high tetracycline resistance are mainly attributed to ease of access as tetracycline can be obtained over-the-counter and besides, it has been used extensively in Ghana for treating other bacterial infections such as eye and sexually transmitted infections in humans [33]. Tetracycline also remains one of the frequently used antibiotics in animal health care in Ghana [34]. However, the genetic basis for tetracycline resistance in Ghana is poorly understood. The *tetM* gene has been found in two independent pneumococcal carriage and invasive disease study in Ghana. However, both studies performed in silico analysis following whole-genome sequencing [8,18]. In contrast, our study amplified the *tetM* gene in all 43 MDR isolates. It was interesting to note that all 43 tetracycline-resistant isolates harbored the *tetM* gene. The two previous studies from Ghana and those from other countries have shown a high detection rate of the *tetM* gene among tetracycline-resistant isolates [8,12,18]. Tetracycline resistance is mediated by the acquisition of *tetM* and *tetO* resistance genes with the latter rarely reported in pneumococci. Further, conjugate transposons such as Tn1545 and Tn916, which can carry other resistance genes including *tetM*, undergo horizontal transfer, which may facilitate the spread of MDR to other isolates [12,35].

In recent years, increased erythromycin resistance <10–>90% has been reported globally with variations in geographical locations, thereby making erythromycin resistance more common than penicillin resistance [13]. In this study, erythromycin resistance wasobserved in 25.6% of our MDR isolates. This prevalence is relatively low compared to reports from other countries [28,36,37], where consumption of erythromycin is relatively high. The low prevalence can be attributed to the in-frequent prescription and use of erythromycin and other macrolides in public hospitals and the communities in Ghana [38]. Although the use of erythromycin is relatively low among the Ghanaian population, it has been shown that the use of other macrolides such as azithromycin could have an impact on pneumococcal resistance [39]. Mass drug administration of azithromycin in selected communities at risk of the yaws disease spells a potential future threat to developing macrolide resistance in pneumococcal strains as seen in Burkina Faso [39,40]. Apart from these factors potentially influencing macrolide resistance, data on the genetic determinants of erythromycin resistance in Ghana are scarce. This study identified the *ermB* and *mefA* genes in 7 (63.6%) and 4 (36.4%), respectively, of the 11 erythromycin-resistant isolates. None of our isolates carried the *mefE* gene and we did not observe co-carriage of the *ermB* and *mefA* genes which is in contrast to other reports [13,36,37]. The *ermB* gene was detected in serotypes 35B, 23B, 6B, and 19F, while the *mefA* gene was limited to serotypes 9V, 38, and 19F. Similarly, two recent studies detected the *ermB* gene in a selection of their isolates without the detection of the *mefA* gene [8,18].

Of the 40 (93% of MDR) penicillin non-susceptible isolates, 26 (65%) showed alterations in their *pbp2b* gene region. We generally observed that penicillin non-susceptibility was predominantly present in serotypes contained in the PCV13. Therefore, we can predict that with the continuous use of PCV13 in Ghana, there could be further reduction in penicillin resistance. Previous studies in Ghana have also observed declining penicillin resistance since the roll-out of PCV13 in Ghana in 2012 [5,8,18]. Our study may not have exhaustively identified the genetic basis of resistance as penicillin non-susceptibility could also result from alterations in the other five penicillin-binding proteins, which we did not study.

The *pcpA* and *psrP* virulence genes were fairly distributed among the identified STs of the MDR isolates. Pili enables the pneumococci to adhere to host epithelial cells, and they are associated with biofilm formation and antibiotic resistance [41]. There are variations in prevalence of the pilus islets reported from different geographical locations [41]. A recent study from Indonesia reported *PI* prevalence of 11.5% [42] in contrast with 33.7 from Iceland [43] and 32.6% from this study [43]. These studies [41,42,43] have shown that isolates carrying the pilus islets are resistant to various antibiotics are covered by the PCVs and limited to a few clonal complexes (CC) including CC320, CC271, CC191, and CC156.

However, possessing both *PI-1* and *PI-2* was found to be a unique property of serotype 19F (ST15459 and ST983) [44] as indicated also in this study. It is interesting to note that isolates that make-up ST156, which was previously described to include serotypes 9V and 14 [45], have quite recently expanded after PCV introduction to encompass many more serotypes such as serotype 19A in the USA, and 11A, 19A, and 24F in Spain [46,47].

In the present study, ST156 was associated with serotype 9V similar to reports from South Africa [48]. ST156 (Spain^9V^-3) is a global clone identified by the PMEN to be involved in the dissemination of penicillin resistance. It appears to be a well-disseminated clone globally. However, this is the first time ST156 has been identified in Ghana and reported to the PubMLST database. Interestingly, no other African country aside South Africa has reported this clone before [24,48]. This means that the spread of this clone has largely been reported outside Africa, although a recent publication has reported its decline in Spain, the country where it was first traced [47].

Furthermore, six out of the seven serotype 23F strains clustered into ST802. However, one more isolate, phenotypically serotype 19F, found itself clustering into ST802. This is an indication of a possible capsular switch from serotype 23F to 19F or vice versa. Capsular switching has been documented in many studies conducted in the post-PCV era [49]. This phenomenon describes the exchange or alteration of a portion of the capsular polysaccharide locus to escape host antibodies. Additionally, a capsular switch may arise from the recombination of different fragment sizes. However, since we did not compare in detail the *cps* of the serotype 23F with that of the 19F included in ST802, we can only infer from the data the possible exchange of *cps* loci within this clonal complex [49]. ST802 was previously identified among serotype 19F and 23F pneumococcal strains in Ghana during the pre-vaccination era [9,15,17]. ST802 was previously reported from other countries [24]. The only serotype 23F, which did not cluster into ST802, belonged to ST2174. ST2174 is a double locus variant (DLV) of ST9090, a major clone associated with serotype 19F described in Ghana during the pre-vaccination era [17].

The two serotype 14 MDR isolates clustered into ST8437, a relatively new clone that has only been reported in Gabon [24]. However, further investigations revealed that this clone is a SLV of (ST9091) a serotype 19F strain that was identified in the Northern part of Ghana in 2011 [24]. This implies that this clone might have disseminated to other parts of the country signifying the role of movement in spreading MDR clones.

The only MDR serotype 6B isolate was associated with ST15461, which is a DLV of the international clone ST273 (Greece^6B^-22). Greece^6B^-22 has been described as a global widespread clone with characteristics of penicillin and erythromycin resistance.

Serotype 19F among the MDR strains appeared to be highly clonal as it is comprised of four different STs (of which two were novel). The two novel STs (15458 and 15459) were DLV and TLV of the hypervirulent ST217 clone, associated with serotype 1 in the 2016 pneumococcal outbreak in Ghana [16]. Again in the early 2000s, ST217 caused a major outbreak that involved Ghana and Burkina Faso [50]. The close association between these novel clones of serotype 19F and ST217 mainly of serotype 1 lineage in Ghana demonstrates the gradual expansion of this clone. The possibility of capsular switching as a result of vaccine pressure and possible recombination of the genetic elements within this clone could be contributors.

## 5. Conclusions

In conclusion, our study describes the molecular characteristics of MDR pneumococci isolated from the nasopharynx of healthy children under five years in Cape Coast, Ghana. The majority of the serotypes were covered by PCV13. Susceptibility to ceftriaxone remained high and hence remains a potential drug of choice in treating pneumococcal infections in Ghana. The presence of the *tetM*, *pbp2b*, *ermB*, and *mefA* genes formed the genetic basis for tetracycline, penicillin, and erythromycin resistance, respectively. Virulence genes *lytA*, *pavB*, *cpsA*, *pcpA*, and *psrP* were fairly distributed among the different serotypes, whereas the pilus islets were limited to serotypes 19F, 6B, and 9V. A robust pneumococcal carriage and invasive disease surveillance system in Ghana is required to monitor internationally disseminating clones such as ST156 (Spain^9V^-3) found in this study. A limitation of this study is the is the fewer number of isolates sequenced by MLST. This limits the extent to which our results can be generalized and compared with other studies.

## Figures and Tables

**Figure 1 microorganisms-10-00469-f001:**
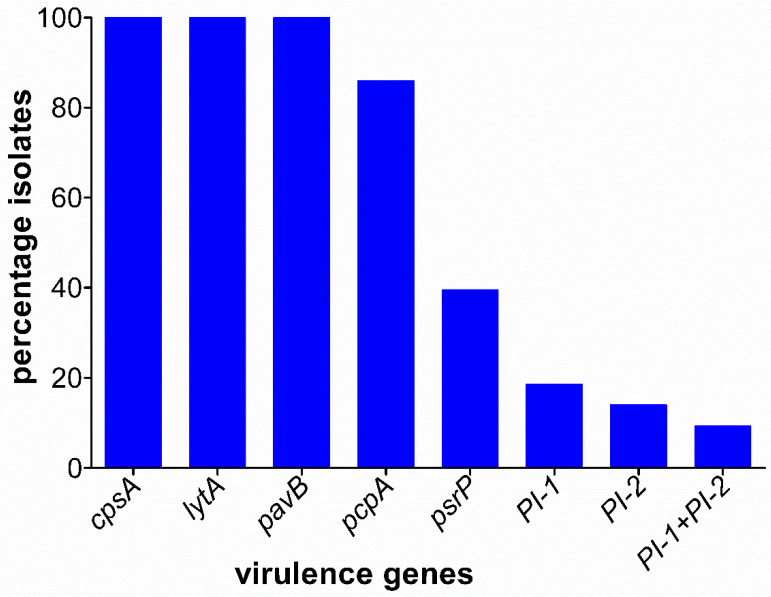
Prevalence of virulence genes among 43 MDR pneumococcal isolates.

**Figure 2 microorganisms-10-00469-f002:**
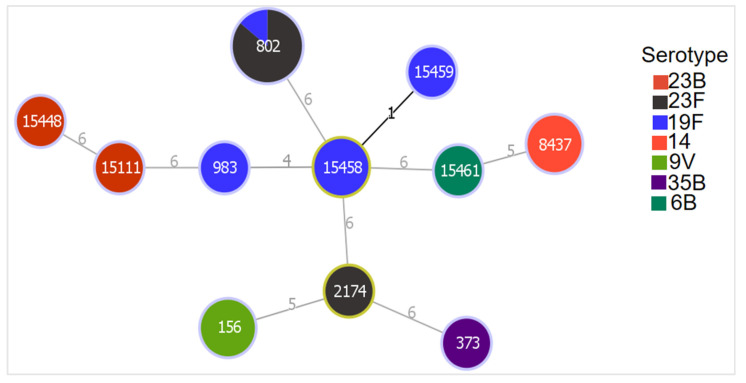
Genetic relatedness between MDR strains of different serotypes. Colors show the presence of different serotypes. Diameters of the nodes are proportional to the number of isolates. Founder STs have a yellow color around their nodes. Branch labels correspond to the number of allelic variations between STs; branch lengths are not to scale.

**Figure 3 microorganisms-10-00469-f003:**
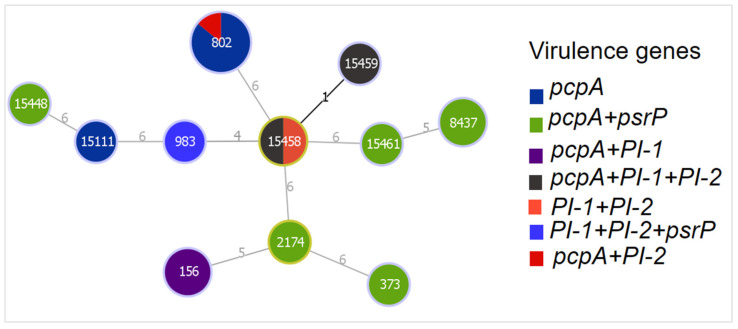
Distribution of virulence genes among STs of MDR strains. Colors show the presence of different virulence genes. Diameters of the nodes are proportional to the number of isolates. Founder STs have a yellow color around their nodes. Branch labels correspond to the number of allelic variations between STs; branch lengths are not to scale.

**Figure 4 microorganisms-10-00469-f004:**
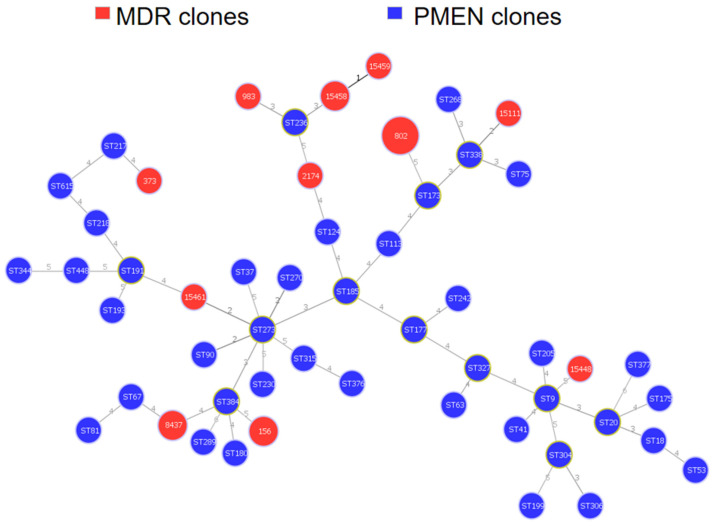
MLST comparison of the genetic relatedness of MDR strains with PMEN clones. The red color represents STs of MDR strains from this study. The blue color shows the PMEN STs obtained from the PMEN database. Diameters of the nodes are proportional to the number of isolates. Founding STs have a yellow color around their nodes. Branch labels correspond to the number of allelic variations between STs; branch lengths are not to scale.

**Table 1 microorganisms-10-00469-t001:** Distributions of MDR serotypes by antibiotic resistance patterns and prevalence of resistance genes.

Pneumococcal Isolates		Antibiotic Resistance Number (%)							Antibiotic Resistance Genes Number (%)			
Serotypes	Number	CRO	ERY	CLI	TET	CHL	COT	PEN	*Pbp2b*	*tetM*	*ermB*	*mefA*
23B	10	0	1(10)	1(10)	10(100)	0(0)	10(100)	10(100)	4(40)	10(100)	1(10)	0(0)
23F	9	0	0.	0	9(100)	7(77.8)	9(100)	9(100)	8(88.8)	9(100)	0(0)	0(0)
19F	6	1(16.7)	4(66.7)	4(66.7)	6(100)	1(16.7)	6(100)	6(100)	6(100)	6(100)	3(50)	1(16.7)
14	4	0	0	0	4(100)	2(50)	4(100)	4(100)	3(75)	4(100)	0(0)	0(0)
6B	3	0	1	1(33.3)	3(100)	0(0)	3(100)	3(100)	2(66.7)	3(100)	1(33.3)	0(0)
15A	3	1(33.3)	0	0	3(100)	1(33.3)	3(100)	1(33.3)	0(0)	3(100)	0(0)	0(0)
35B	2	0	2(100)	2(100)	2(100)	0(0)	2(100)	1(50)	0(0)	2(100)	2(100)	0(0)
3	2	0	0	0	2(100)	0(0)	2(100)	2(100)	1(50)	2(100)	0(0)	0(0)
9V	2	0	2(100)	0	2(100)	0(0)	2(100)	2(100)	2(100)	2(100)	0(0)	2(100)
6A	1	0	0	0	1(100)	0(0)	1(100)	1(100)	0(0)	1(100)	0(0)	0(0)
38	1	0	1(100)	0	1(100)	0(0)	1(100)	1(100)	0(0)	1(100)	0(0)	1(100)

CRO, ceftriaxone; ERY, erythromycin; CLI, clindamycin; TET, tetracycline; CHL, chloramphenicol; COT, cotrimoxazole; PEN, penicillin.

**Table 2 microorganisms-10-00469-t002:** MLST of 20 selected multidrug-resistant pneumococci isolates.

Isolate ID	*aroE*	*gdh*	*gki*	*recP*	*spi*	*xpt*	*ddl*	ST	Serotype
S125	7	11	10	1	6	8	1	156	9V
S276	7	11	10	1	6	8	1	156	9V
S106	7	13	4	5	7	88	9	373	35B
H130	10	13	53	1	72	38	31	802	19F
H148	10	13	53	1	72	38	31	802	23F
S33	10	13	53	1	72	38	31	802	23F
S34	10	13	53	1	72	38	31	802	23F
S305	10	13	53	1	72	38	31	802	23F
S341	10	13	53	1	72	38	31	802	23F
S578	10	13	53	1	72	38	31	802	23F
S243	15	16	19	15	3	104	63	983	19F
S41	7	16	8	8	6	142	14	2174	23F
S294	2	89	9	38	6	1	18	8437	14
S300	2	89	9	38	6	1	18	8437	14
H9	4	16	19	15	55	20	31	**15458 ***	19F
S26	4	16	19	15	55	16	31	**15459 ***	19F
S85	4	16	19	15	55	20	31	**15458 ***	19F
S579	8	6	1	2	6	1	31	**15461 ***	6B
S237	1	43	41	18	13	37	8	**15448 ***	23B
S238	12	13	8	6	3	6	8	**15111 ***	23B

* Novel STs are in bold font.

## Data Availability

All data generated from this study can be made available by the authors upon request. In addition, all MLST data are publicly available in the PubMLST database.

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
