# Peer review of "Molecular Epidemiology of Multidrug-Resistant Pneumococci among Ghanaian Children under Five Years Post PCV13 Using MLST"

_microorganisms, 2022, doi:10.3390/microorganisms10020469_

Round 1
Reviewer 1 Report
The authors further characterize pneumococcal isolates previously collected and described in Reference #5 from a carriage study in healthy children in Ghana in February 2018. This paper focuses only on a subset of isolates found to be multi-drug resistant with the aim of describing the resistance and virulence determinants. While a large majority of this data seems to overlap with that already published in ref #5 this does provide some new information on pneumococcal lineages and mechanisms of resistance among isolates from Ghana, where data is lacking.
Specific comments for improvement of the manuscript are below:
Introduction, page 1, line 40-41. The WHO report referenced (ref 1) refers to organisms identified with concerns around antimicrobial resistance. You should mention that the pneumococcus is included in this report due to antibiotic resistance.
Introduction, page 3, line 98. Please can you provide reference for the global antibiotic resistance meta-data analysis that you mention showing no impact of PCV on tetracycline, erythromycin and cotrimoxazole resistance.
Materials and Methods, page 3, line 126. I assume penicillin non-susceptible should be 0.12 – 1 µg/ml?
Materials and Methods, page 4, lines 158-159. Do you mean >= 4 different classes of antibiotics? I am confused as to why you chose only to do MLST on 20 isolates meeting this criteria when the purpose of the study was to characterize MDR strains resistant to >=3 antibiotics? Why narrow to so few isolates and not test all 43. Your numbers are already very small with 43 and with 20 this makes it very difficult to make significant conclusions.
Results, page 4, line 195-196. Here I would suggest including the total number of kids enrolled in the study ie. 151 isolates were obtained from 513 children < 5 years of age, of which 43 were MDR.
Results, page 5, line 203. Would suggest replacing “highly” with “fully”
Results, page 5, lines 203-208. I don’t see any reference to penicillin intermediate and resistant results? Can you include how many isolates had MICs between 0.12-1ug/ml and how many were considered fully resistant. Or were all 40/43 (93%) considered resistant?
Results, page 5, lines 214-215. I assume that the alterations in the pbp2b is referring to samples that were PCR positive using the assay from du Plessis et al? Interesting that your Etest results show 93% resistant but that the pbp2b PCR that is supposed to determine penicillin resistance was only positive in 26 (65%) of isolates. Can you explain this discordance?
Results, page 5, lines 225-228. The presence of pili among these isolates which are MDR seems low. Can you compare and discuss with other published data on pili prevalence among antibiotic-resistant strains.
Fig 2. I believe that the colors show serotype? Not different virulence genes?
Page 9, line 302-303. Tetracycline has been used extensively in Ghana for what? Treatment of certain infections? Please clarify/elaborate and provide a reference to support this claim.
Page 9, lines 304-305. I don’t entirely agree with this statement from 2 previous studies. Ref#8 tested 12 isolates of which 11 were tetM positive and resistant to tetracycline? Ref# 18 tested 26 isolates by WGS and while they did not specifically mention resistance determinant, found a high proportion of isolates also resistant to tetracycline. In your study it is not surprising that you found tetracycline resistance in all 43 since you focused only on MDR isolates. Also, by far globally, tetM is the major resistance determinant identified among tetracycline resistant pneumococci with tetO being very rare. Nothing new or novel here.
Page 10, lines 354-355. The MLST database cannot be used as main source for determining where specific STs are located globally. The MLST does not capture this fully and I would err on the cautious side using this to make this statement about ST156. ST156 has also previously been identified in North and South America in many studies so I don’t think you can say it is largely limited to Europe. Please rephrase.
Page 10, line 359-360. Or could the switch be from 19F to 23F with expansion of 23F ST802 lineage?
Page 11, lines 400-403. This study describes carriage from healthy children. I would envision that a robust surveillance system would need to include IPD?
Author Response
Reviewer 1
The authors further characterize pneumococcal isolates previously collected and described in Reference #5 from a carriage study in healthy children in Ghana in February 2018. This paper focuses only on a subset of isolates found to be multi-drug resistant with the aim of describing the resistance and virulence determinants. While a large majority of this data seems to overlap with that already published in ref #5 this does provide some new information on pneumococcal lineages and mechanisms of resistance among isolates from Ghana, where data is lacking.
Specific comments for improvement of the manuscript are below:
Introduction, page 1, line 40-41. The WHO report referenced (ref 1) refers to organisms identified with concerns around antimicrobial resistance. You should mention that the pneumococcus is included in this report due to antibiotic resistance.
Response: We thank the reviewer for pointing this out. The statement has been modified.
please see line 41: ‘As such, the pneumococcus is included in the nine bacterial pathogens of international concern mainly due to the detection of antibiotic resistance particularly penicillin-non-susceptibility‘
Introduction, page 3, line 98. Please can you provide reference for the global antibiotic resistance meta-data analysis that you mention showing no impact of PCV on tetracycline, erythromycin and cotrimoxazole resistance.
Response: The reference to the above statement has been provided, please see line 106.
Materials and Methods, page 3, line 126. I assume penicillin non-susceptible should be 0.12 – 1 µg/ml?
Response: Yes, this has been corrected, thank you. Please see line 135.
Materials and Methods, page 4, lines 158-159. Do you mean >= 4 different classes of antibiotics? I am confused as to why you chose only to do MLST on 20 isolates meeting this criteria when the purpose of the study was to characterize MDR strains resistant to >=3 antibiotics? Why narrow to so few isolates and not test all 43. Your numbers are already very small with 43 and with 20 this makes it very difficult to make significant conclusions.
Response: We agree with the reviewer regarding the fewer number of isolates sequenced. However, this was the result of financial constraints. The isolates had to be limited to 20 isolates (i.e., isolates resistant to ≥ 4 antibiotics) because they were being sequenced in addition to other isolates for which the cost of sequencing was very high. We therefore consider this as a limitation of our study and have included a statement in the manuscript. For your knowledge: the 1st author Richael Mills conducted this project as a DAAD (German Academic Exchange Service) –fellowship without any budget for consumables. Please see line 438-440.
Results, page 4, line 195-196. Here I would suggest including the total number of kids enrolled in the study ie. 151 isolates were obtained from 513 children < 5 years of age, of which 43 were MDR.
Response: We thank the reviewer for this suggestion. The statement has been revised. Please see line 205-206.
Results, page 5, line 203. Would suggest replacing “highly” with “fully”
Response: Kindly see line 213, the word “highly” has been replaced with “fully”
Results, page 5, lines 203-208. I don’t see any reference to penicillin intermediate and resistant results? Can you include how many isolates had MICs between 0.12-1ug/ml and how many were considered fully resistant. Or were all 40/43 (93%) considered resistant?
Response: We thank the reviewer for pointing this out. Of the 43 isolates, 39 showed intermediate (MICs 0.12-1.0 μg/ml) resistance, while one isolate was considered fully resistant (≥2 μg/ml). We have included a statement on this in the results, please see line 217.
Results, page 5, lines 214-215. I assume that the alterations in the pbp2b is referring to samples that were PCR positive using the assay from du Plessis et al? Interesting that your Etest results show 93% resistant but that the pbp2b PCR that is supposed to determine penicillin resistance was only positive in 26 (65%) of isolates. Can you explain this discordance?
Response:
Indeed, the assay from du Plessis et al was used for the determination of alterations in the pbp2b gene region.
The discordance between the Etest and pbp2b PCR results is expected as there are other penicillin binding proteins such as pbp1a, pbp2a and pbp2x, which also confer penicillin resistance.
In this study we only focused on the pbp2b gene, which has been shown by du Plessis et al., to be very diverse and hence alterations in this gene region contributes significantly to penicillin resistance.
Therefore, we stated in line 369-372 that our study may not have thoroughly identified the genetic basis of penicillin resistance.
Results, page 5, lines 225-228. The presence of pili among these isolates which are MDR seems low. Can you compare and discuss with other published data on pili prevalence among antibiotic-resistant strains.
Response: A brief discussion on the pili prevalence and antibiotic resistance has been included in the discussion. Please see line 373-380.
Fig 2. I believe that the colors show serotype? Not different virulence genes?
Response: We thank the reviewer for pointing this out. The correction has been made. Please see line 288.
Page 9, line 302-303. Tetracycline has been used extensively in Ghana for what? Treatment of certain infections? Please clarify/elaborate and provide a reference to support this claim.
Response: A supporting statement on the extensive use of tetracycline in Ghana has been provided. Please see line 330-334
Page 9, lines 304-305. I don’t entirely agree with this statement from 2 previous studies. Ref#8 tested 12 isolates of which 11 were tetM positive and resistant to tetracycline? Ref# 18 tested 26 isolates by WGS and while they did not specifically mention resistance determinant, found a high proportion of isolates also resistant to tetracycline. In your study it is not surprising that you found tetracycline resistance in all 43 since you focused only on MDR isolates. Also, by far globally, tetM is the major resistance determinant identified among tetracycline resistant pneumococci with tetO being very rare. Nothing new or novel here.
Response: Please see line 333-334, the above statement has been revised.
Page 10, lines 354-355. The MLST database cannot be used as main source for determining where specific STs are located globally. The MLST does not capture this fully and I would err on the cautious side using this to make this statement about ST156. ST156 has also previously been identified in North and South America in many studies so I don’t think you can say it is largely limited to Europe. Please rephrase.
Response: Please see line 392, the statement has been revised.
Page 10, line 359-360. Or could the switch be from 19F to 23F with expansion of 23F ST802 lineage?
Response: Indeed, a switch from 19F to 23F is also possible. The statement has been revised to capture this possibility. Please see line 396.
Page 11, lines 400-403. This study describes carriage from healthy children. I would envision that a robust surveillance system would need to include IPD?
Response: We agree with the reviewer that an IPD surveillance system is crucial in monitoring the changing epidemiology of the pneumococcus in Ghana
Reviewer 2 Report
Mills RO et al. “Molecular epidemiology of multidrug-resistant pneumococci among Ghanaian children under five years post PCV13 using MLST”
In this study, the authors described antibiotic resistance, prevalence of serotype and virulence genes, genetic epidemiology of S. pneumoniae from Ghanaian children.
I would like to point out a few questionable points.
- Why is serotyping performed on 43 MDR isolates, but MLST only on 20 isolates?
I think that serotyping, antibiotic susceptibility testing, and MLST should be performed on all isolates (153 isolates) to satisfy the purpose of this study. If there is not analysis of entire isolates, the logic of the discussion in this paper is greatly weakened.
- Line 126. Penicillin resistance breakpoint is ≥8 mg/L according the CLSI.
- Line 138. I don’t know the meaning of “characterization of MDR isolates”. DNA extraction is not characterization. What is the purpose of DNA extraction?
- Line 157. The procedure of MLST is quite standard. Thus, this part can be shortened.
- Line 223 and Figure 1. Analysis of which virulence gene is present in how many isolates is not very meaningful. Instead, it is more meaningful to analyze which virulence gene each genotype or serotype has.
- Figures 2 and 3 are not necessary.
Author Response
Reviewer 2
In this study, the authors described antibiotic resistance, prevalence of serotype and virulence genes, genetic epidemiology of S. pneumoniae from Ghanaian children.
I would like to point out a few questionable points.
- Why is serotyping performed on 43 MDR isolates, but MLST only on 20 isolates?
I think that serotyping, antibiotic susceptibility testing, and MLST should be performed on all isolates (153 isolates) to satisfy the purpose of this study. If there is not analysis of entire isolates, the logic of the discussion in this paper is greatly weakened.
Response:
We thank the reviewer for pointing out this observation. Indeed, sequencing all 151 isolates isolated from the 513 healthy children would provide detailed information on the clonality of our pneumococcal collection in comparison with isolates from other countries. We plan to source for funds to sequence the rest of the isolates to share this data with the pneumococcal community. Nonetheless, we have stated that the small number of isolates sequenced in this study is as a result of high sequencing cost which could not fit well with our research budget for this project, which the 1st author Richael Mills conducted as a DAAD (German Academic Exchange Service) –fellowship, without any budget for consumables. Please see line 438-439, where we made a statement of the limitation of the study.
- Line 126. Penicillin resistance breakpoint is ≥8 mg/L according the CLSI.
Response: We have used the MIC breakpoints for oral penicillin V in which the penicillin resistance breakpoint is ≥2 μg/ml according to the CLSI guidelines (line 135).
- Line 138. I don’t know the meaning of “characterization of MDR isolates”. DNA extraction is not characterization. What is the purpose of DNA extraction?
Response: We thank the reviewer for pointing this out. The statement has been revised. Please see line 147
- Line 157. The procedure of MLST is quite standard. Thus, this part can be shortened.
Response: In as much as we agree with the reviewer, we also believe that this section contains other vital information that gives a vivid description of how the MLST procedure was performed, how the sequencing results were analyzed, and which analytical software were used.
- Line 223 and Figure 1. Analysis of which virulence gene is present in how many isolates is not very meaningful. Instead, it is more meaningful to analyze which virulence gene each genotype or serotype has.
Response: Even though Figure 1 may seem not to carry much meaning, it shows in simple terms the prevalence of the various virulence genes present in our MDR isolates. Further, the relationship between virulence genes and serotypes and genotypes are shown in Figures 2 and 3.
- Figures 2 and 3 are not necessary.
Response: Figure 2 shows immediately how the distribution of the different serotypes among the identified sequence types (STs) and how they are related. While Figure 3 depicts the pattern of virulence genes in the different STs giving a clear indication of how these genes are associated with different STs identified in this study.
Reviewer 3 Report
The manuscript brings a very interesting and relevant topic but it has some points that need to be improved.
1) Line 93- One of the expected benefits of PCV introduction is to reduce antibiotic resistance 93 including MDR
Vaccination programs are implemented to reduce mortality and severity of infection, not the development of antimicrobial resistance. Do the authors have any reliable references that justify this expectation? Please review this point.
2) Describe serotyping appropriately in the methodology with the methods cited in the abstract.
3) Table 1- improve table format and cell alignment horizontally and vertically.
4) Considering the affirmation that follow, please clarify whether all antimicrobials can be purchased over-the-counter in Ghana or just tetracycline – Lines – 299-303.
“In Ghana, persistently high resistance to tetracycline has been seen in pneumococci and other bacteria [11]. Factors fueling the high tetracycline resistance are mainly attributed to ease of access as tetracycline can be obtained over-the-counter and besides it has been used extensively in Ghana”
5) At Conclusion- Line 395- “The majority of the serotypes were of PCV13 origin”
The above statement may give the impression that the vaccine is spreading strains of these serotypes and that it have resistance characteristics. This fact cannot occur since vaccines are bacterin, but the incorrect idea can cause confusion about the safety of vaccination.
Please review the way of writing so that the idea to be expressed is clearer.
Author Response
Reviewer 3
The manuscript brings a very interesting and relevant topic but it has some points that need to be improved.
1) Line 93- One of the expected benefits of PCV introduction is to reduce antibiotic resistance 93 including MDR
Vaccination programs are implemented to reduce mortality and severity of infection, not the development of antimicrobial resistance. Do the authors have any reliable references that justify this expectation? Please review this point.
Response: The point on expected benefits of PCV introduction have been revised. Please see line 95-101
2) Describe serotyping appropriately in the methodology with the methods cited in the abstract.
Response: We thank the reviewer for the suggestion. We have included a brief statement on the methods used for the identification and serotyping of the isolates. Please see lines 118-120.
3) Table 1- improve table format and cell alignment horizontally and vertically.
Response: Table 1 has been formatted please see line 240
4) Considering the affirmation that follow, please clarify whether all antimicrobials can be purchased over-the-counter in Ghana or just tetracycline – Lines – 299-303.
“In Ghana, persistently high resistance to tetracycline has been seen in pneumococci and other bacteria [11]. Factors fueling the high tetracycline resistance are mainly attributed to ease of access as tetracycline can be obtained over-the-counter and besides it has been used extensively in Ghana”
Response: Not all antimicrobials can be obtained over the counter in Ghana. However, common antibiotics that can be administered orally or topically such as tetracycline, cotrimoxazole, amoxicillin can be obtained over the counter. Further information has been provided on the extensive use of tetracycline in Ghana. Please see lines 331-334.
5) At Conclusion- Line 395- “The majority of the serotypes were of PCV13 origin”
The above statement may give the impression that the vaccine is spreading strains of these serotypes and that it have resistance characteristics. This fact cannot occur since vaccines are bacterin, but the incorrect idea can cause confusion about the safety of vaccination.
Please review the way of writing so that the idea to be expressed is clearer.
Response: The authors have also reviewed the manuscript again to ensure the entire content is clear, for easy reading and understanding. We thank the reviewer for this suggestion.
Round 2
Reviewer 2 Report
Regarding the answer to the first point,
I was not talking about whole genome sequencing of whole strains, I was referring to serotyping and MLST. This is less expensive than WGS for the whole strain, and I think it corresponds to the basic information about the strains.
Author Response
Regarding the answer to the first point,
I was not talking about whole genome sequencing of whole strains, I was referring to serotyping and MLST. This is less expensive than WGS for the whole strain, and I think it corresponds to the basic information about the strains.
Response:
This seems to be a misunderstanding and we apologize for this.
In fact, serotyping (by multiplex-PCR and Quellung reaction) and antibiotic susceptibility testing has been done for all 151 pneumococcal isolates. This was published in doi: 10.3390/microorganisms8121987 and the paper has been cited several times in the submitted manuscript. For MLST we concentrated in the submitted study on isolates resistant to more than 4 different classes of antibiotics, meaning the most interesting isolates among all isolates. Of course, knowlegde of the ST or CC of the other isolates is interesting, but does not add much to the submitted study, because the main issue here is the characterization of MDR strains. In fact, the results show that within this group are 6 novel STs (Table 2), indicating that the selection of MDR strains was sufficient.